# Computing the Parameter Values for the Emergence of Homochirality in Complex Networks

**DOI:** 10.3390/life9030074

**Published:** 2019-09-15

**Authors:** Andrés Montoya, Elkin Cruz, Jesús Ágreda

**Affiliations:** 1Departamento de Matemáticas, Universidad Nacional de Colombia, Bogotá D. C. 111321, Colombia; jamontoyaa@unal.edu.co; 2Departamento de Química, Universidad Nacional de Colombia, Bogotá D. C. 111321, Colombia; elacruzca@unal.edu.co

**Keywords:** models of biological homochirality, mirror symmetry-breaking, algorithmic problems, semialgebraic definitions, stoichiometric network analysis

## Abstract

The goal of our research is the development of algorithmic tools for the analysis of chemical reaction networks proposed as models of biological homochirality. We focus on two algorithmic problems: detecting whether or not a chemical mechanism admits mirror symmetry-breaking; and, given one of those networks as input, sampling the set of racemic steady states that can produce mirror symmetry-breaking. Algorithmic solutions to those two problems will allow us to compute the parameter values for the emergence of homochirality. We found a mathematical criterion for the occurrence of mirror symmetry-breaking. This criterion allows us to compute semialgebraic definitions of the sets of racemic steady states that produce homochirality. Although those semialgebraic definitions can be processed algorithmically, the algorithmic analysis of them becomes unfeasible in most cases, given the nonlinear character of those definitions. We use Clarke’s system of convex coordinates to linearize, as much as possible, those semialgebraic definitions. As a result of this work, we get an efficient algorithm that solves both algorithmic problems for networks containing only one enantiomeric pair and a heuristic algorithm that can be used in the general case, with two or more enantiomeric pairs.

## 1. Introduction

We study mathematical models of absolute asymmetric synthesis [1,2] that are used to explain the emergence of biological homochirality, which is, according to Frank [3], a natural property of life. Frank proposed, as early as 1953, a chemical mechanism to support his thesis. An important feature of this minimal model is that homochirality is produced by dynamic instability [4]. After Frank many other more sophisticated networks have been proposed (see for instance [5,6,7]), but all of them are based on the same idea: homochirality is the product of chemical instabilities. It is important to remark that the mathematical (stability) analysis [8] of those complex models is a hard piece of work. Fortunately, we found a particular symmetry in the Jacobian matrices of those models [9] that yields semialgebraic definitions of the instability regions where the symmetry-breaking can be observed. Most of those semialgebraic expressions are highly nonlinear and hard to sample. We used Clarke’s Stoichiometric Network Analysis (SNA) [10] to reduce the complexity of those expressions.

All those ingredients were put together into an algorithmic tool, and software Listanalchem [11], that can be used to test models proposed to explain the origin of homochirality, and which can also help us to build new and better models. Further thermodynamics constraints must be taken into account, but that point will not be discussed here.

We begin with a mathematical presentation of the method. After that, we use the developed algorithm to analyze three representative models of biological homochirality taken from the available literature.

## 2. Network Models of Absolute Asymmetric Synthesis

We use the term *absolute asymmetric synthesis* to designate all the possible chemical mechanisms that, operating in achiral environments, can transform a racemic mixture into an enantiopure one. We suppose that all those chemical processes reduce to finite sets of chemical reactions acting on finite sets of chemical species. Thus, we suppose that all those processes can be suitably described by *chemical reaction networks*.

**Definition** **1.**
*A chemical reaction over the chemical species $X_{1},\ldots ,X_{n}$ is an expression like*
(1)$$c_{1}X_{{}_{1}}+\cdots +c_{n}X_{n}\to d_{1}X_{1}+\cdots +d_{n}X_{n},$$
*where $c_{1},\ldots ,c_{n}$ and $d_{1},\ldots ,d_{n}$ are small integers (some of which could be equal to zero). The latter expression indicates that the mixture of $c_{1}$ units of $X_{1},\ldots ,$ and $c_{n}$ units of $X_{n}$ gives place to $d_{1}$ units of $X_{{}_{1}},\ldots ,$ and $d_{n}$ units of $X_{n}.$*

*A chemical reaction network over the species $\left\{X_{1},\ldots ,X_{n}\right\}$ is a set of chemical reactions, say the set $\left\{R_{1},\ldots ,R_{r}\right\},$ over this set of species.*


**Notation** **1.**
*Given a chemical network $\Omega =\left\{\left(X_{1},\ldots ,X_{n}\right),\left(R_{1},\ldots ,R_{r}\right)\right\}$, we use the expression*
(2)$$c_{1i}X_{{}_{1}}+\cdots +c_{ni}X_{{}_{n}}\to d_{1i}X_{{}_{1}}+\cdots +d_{ni}X_{n}$$
*to denote the reaction $R_{i},$ and we use the symbol $k_{i}$ to denote its reaction rate constant. We use variables $\left[X_{1}\right],\ldots ,\left[X_{n}\right]$ to denote the concentrations of the n chemical species.*


Let us consider an example of a chemical reaction network. *Frank network* is the network $\Omega_{F}=\left\{\left(L,D,A,P\right);\left(R_{1},R_{2},R_{3}\right)\right\}$, where:

reaction $R_{1}$:L + A $\underset{\to }{k_{1}}$ 2L,reaction $R_{2}$:D + A $\underset{\to }{k_{2}}$ 2D, andreaction $R_{3}$:L + D $\underset{\to }{k_{3}}$ P.

**Remark** **1.**
*It is important to remark that the network $\Omega_{F}$ was the first, and it is the most elementary model of absolute asymmetric synthesis proposed in the literature [3].*


The dynamics of a network Ω reduces to the temporal evolution of the *concentration variables*
$\left[X_{1}\right],\ldots ,\left[X_{n}\right].$ We suppose that those temporal evolutions are completely determined by the *law of mass action*[12], that is: we suppose that the temporal evolution of the variables $\left[X_{1}\right],\ldots ,\left[X_{n}\right]$ is given by the system of Ordinary Differential Equations (ODE)
(3)$$\frac{d\left[X_{i}\right]}{dt}=\sum_{j=1}^{r}k_{j}\left(d_{ij}-c_{ij}\right)\left({\left[X_{1}\right]}^{c_{1j}}\cdot \cdots \cdot {\left[X_{n}\right]}^{c_{nj}}\right),\;i=1,\ldots ,n,$$
where given j⩽r the symbol $k_{j}$ denotes the reaction rate constant of $R_{j}$. We get consequently that all those dynamics are deterministic: their evolution in time is entirely determined by their initial states.

**Remark** **2.**
*Let s be a (initial) state of network Ω, state s is a vector that encodes the values of all the parameters that participate in the dynamics of Ω. Thus, we have that s can be fully described by a (n+r)-tuple*
(4)$$\left(\left[X_{1}\right],\ldots ,\left[X_{n}\right],k_{1},\ldots ,k_{r}\right),$$
*of nonnegative reals. We must notice that the entries $k_{1},\ldots ,k_{r}$ could remain constant along the dynamics, but in despite this, we choose to include the reaction rate constants in our notion of state.*


**Definition** **2.**
*A state $\left(\left[X_{1}\right],\ldots ,\left[X_{n}\right],k_{1},\ldots ,k_{r}\right)$ is said to be a steady state, if and only if, it satisfies the following system of polynomial equations:*
(5)$$\begin{array}{cc} 0=\frac{d\left[X_{1}\right]}{dt} & =\sum_{i=1}^{r}\left(d_{1i}-c_{1i}\right)k_{i}{\left[X_{1}\right]}^{c_{1i}}\cdots {\left[X_{n}\right]}^{c_{ni}},\\ \; & \vdots \end{array}$$
(6)$$\begin{array}{cc} 0=\frac{d\left[X_{n}\right]}{dt} & =\sum_{i=1}^{r}\left(d_{ni}-c_{ni}\right)k_{i}{\left[X_{1}\right]}^{c_{1i}}\cdots {\left[X_{n}\right]}^{c_{ni}}. \end{array}$$


Notice that the steady states are the mathematical equilibria of the system. However, if one thinks in chemical equilibrium, it could be possible to consider the states for which only the set of forward reaction rates vanish. Also, it could be possible to consider the existence of complex networks containing loops, where not all forward rates vanish identically at the steady state.

Let us consider the case of network $\Omega_{F}$ when the reagent $\left[A\right]$ is assumed constant, according to the pool chemical approximation, see [13] chapters 2 and 3. The dynamics of this network is governed by the ODE system
(7)$$\begin{array}{ccc} \frac{d\left[L\right]}{dt} & = & k_{1}\left[L\right]\left[A\right]-k_{3}\left[L\right]\left[D\right], \end{array}$$
(8)$$\begin{array}{ccc} \frac{d\left[D\right]}{dt} & = & k_{2}\left[D\right]\left[A\right]-k_{3}\left[L\right]\left[D\right]. \end{array}$$

The states of $\Omega_{F}$ are septuples $\left(\left[L\right],\left[D\right],\left[A\right],\left[P\right],k_{1},k_{2},k_{3}\right)$ of nonnegative reals, and we have that such septuples encode racemic steady states, if and only if, the equality
(9)$$\begin{array}{ccc} 0 & = & k_{1}\left[L\right]\left[A\right]-k_{3}\left[L\right]\left[D\right]=k_{2}\left[D\right]\left[A\right]-k_{3}\left[L\right]\left[D\right] \end{array}$$
hold. Observe that $k_{1}$ must be equal to $k_{2}$; otherwise, the network encodes a chiral environment. This means $\left[L\right]=\left[D\right]>0$. It is important to remark, at this point, that any physical realization of Frank’s model implies a system open to matter flow to maintain [A] constant. Open systems used to have larger and more complex sets of steady states.

Analyzing the dynamics of $\Omega_{F}$ corresponds to analyzing the set $\mathrm{SS}\left(\Omega_{F}\right)$ constituted by all its steady states, as well as the dynamics that can be triggered by arbitrary small perturbations of those states.

### The Goal

We want to contribute to the investigation on the emergence of *biological homochirality* by providing the interested researchers with mathematical and algorithmic tools that can be used in the analysis of any network model of absolute asymmetric synthesis.

**Definition** **3.**
*We say that a chemical reaction network Ω exhibits chiral amplification, if and only if, it has the ability of transforming negligible gaps between the concentrations of the chiral species into larger gaps.*


We would like to characterize the set of chemical reaction networks that exhibit chiral amplification.

Let $\Omega =\left\{\left(X_{1},\ldots ,X_{N}\right);\left(R_{1},\ldots ,R_{r}\right)\right\}$ be a chemical reaction network, and suppose that $X_{1}$ is the -form of a chiral biomolecule, and that $X_{2}$ is the -form of the same molecule. Suppose that the system got stuck at a steady state satisfying the equality $\left[X_{1}\right]=\left[X_{2}\right]$. There are physical mechanisms that may create small *enantiomeric gaps* (that could perturb this racemic steady state). Those perturbations trigger dynamics, and those latter dynamics could:Evolve towards the original steady states, if, for instance, those states are *stable*.Evolve towards different racemic states, if, for instance, those sets of racemic states are *attractors* of the dynamics.Evolve towards states with large enantiomeric gaps (also *called scalemic states*).

We are interested in the latter case, and we say that in that case, the system undergoes *homochiral dynamics*. We want to characterize the steady states of Ω that can undergo homochiral dynamics. We introduce, below, a precise formulation of the algorithmic problem that we study (and solve) in this paper.

## 3. Pseudochiral Networks

Let $\Omega =\left(X,R\right)$ be a chemical reaction network, and suppose that it is a network model of absolute asymmetric synthesis. Then, the set X must be constituted by three disjoint sets of chemical species, the sets $\left\{L_{1},\ldots ,L_{k}\right\}$; $\left\{D_{1},\ldots ,D_{k}\right\}$ and $\left\{X_{2k+1},\ldots ,X_{n}\right\}$.

The set $\left\{L_{1},\ldots ,L_{k}\right\}$ constitutes the l-side of Ω, the set $\left\{D_{1},\ldots ,D_{k}\right\}$ constitutes the d-side and the set $\left\{X_{2k+1},\ldots ,X_{n}\right\}$ is constituted by all the achiral species occurring in Ω. Moreover, given i≤k we have that $\left(L_{i},D_{i}\right)$ is an *enantiomeric pair*, that is: $L_{i}$ is the -form of a chiral molecule, while $D_{i}$ is the -form of the same molecule.

**Definition** **4.***Let s be a state of* Ω*, and suppose that*
(10)$$s=\left(\left[L_{1}\right],\ldots ,\left[L_{k}\right],\left[D_{1}\right],\ldots ,\left[D_{k}\right],\left[X_{2k+1}\right],\ldots ,\left[X_{n}\right],k_{1},\ldots ,k_{r}\right).$$
*The enantiomeric gap of s is equal to $\sum_{i\leq k}\left|\left[L_{i}\right]-\left[D_{i}\right]\right|.$*


A state of Ω is said to be *racemic*, if and only if, its enantiomeric gap is equal to 0. Thus, the *racemic condition* for Ω corresponds to the following set of polynomial equalities
(11)$$\left[L_{1}\right]=\left[D_{1}\right],\;\left[L_{2}\right]=\left[D_{2}\right],\ldots ,\;\left[L_{k}\right]=\left[D_{k}\right].$$

Recall that we are interested in the racemic states of Ω that can produce homochiral dynamics. It is important to take into account the *indiscernibility of enantiomeric pairs*: it is known that the two species of an enantiomeric pair react with the same achiral species at the same reaction rates. The indiscernibility of enantiomeric pairs implies that any feasible network model of absolute asymmetric synthesis must be a pseudochiral network, as defined below.

**Definition** **5.***Let $\Omega =\left(X,R\right)$ be a network model of absolute asymmetric synthesis, and suppose that*(12)$$X=\left\{L_{1},\ldots ,L_{k},D_{1},\ldots ,D_{k},X_{2k+1},\ldots ,X_{n}\right\},$$*where $\left\{L_{1},\ldots ,L_{k}\right\}$ is the*l*-side of* Ω*, and $\left\{D_{1},\ldots ,D_{k}\right\}$ is its*
d*-side. We say that* Ω *is a pseudochiral network, if and only if, it is indistinguishable from the network that is obtained when one switches the*
l*-species and the*
d*-species.*

Consider the following example of a pseudochiral network. Melvin Calvin, whose work laid the foundations of *chemical evolution*, proposed an abstract model of biological homochirality [14]. Calvin’s mechanism can be suitably described by a chemical reaction network that we denote with the symbol $\Omega_{C}$, and which is defined as follows:

The constituent species of $\Omega_{C}$ are the species $L_{1},L_{2},D_{1}$ and $D_{2}.$ The chemical reactions in $\Omega_{C}$ are:A pair of*autocatalytic reactions*
(13)$$R_{0}:\;L_{1}+L_{2}\;\underset{\to }{k_{0}}2L_{2}\;\;\;\mathrm{and}\;\;\;R_{1}:\;D_{1}+D_{2}\;\underset{\to }{k_{1}}2D_{2},$$
as well as the reverse reactions
(14)$$R_{2}:\;2L_{2}\;\underset{\to }{k_{2}}L_{1}+L_{2}\;\;\;\mathrm{and}\;\;\;R_{3}:\;2D_{2}\;\underset{\to }{k_{3}}D_{1}+D_{2}.$$The *racemization* reactions
(15)$$R_{4}:\;L_{1}\;\underset{\to }{k_{4}}D_{1}\;\;\;\mathrm{and}\;\;\;R_{5}:\;D_{1}\;\underset{\to }{k_{5}}L_{1}.$$The *enantiomeric conversion* reactions
(16)$$R_{6}:\;L_{1}\;\underset{\to }{k_{6}}L_{2}\;\;\;\mathrm{and}\;\;\;R_{7}:\;D_{1}\;\underset{\to }{k_{7}}D_{2},$$
as well as their reverse reactions
(17)$$R_{8}:\;L_{2}\;\underset{\to }{k_{8}}L_{1}\;\;\;\mathrm{and}\;\;\;R_{9}:\;D_{2}\;\underset{\to }{k_{9}}D_{1}.$$

Notice that the reactions in $\Omega_{C}$ are organized in pairs: for each reaction involving the species of the l-side there is a *dual reaction* involving the species of the d-side. We have, for instance, that if the reaction $L_{1}+L_{2}\to 2L_{2}$ occurs, then the dual reaction $D_{1}+D_{2}\to 2D_{2}$ must also occur; otherwise, the network would distinguish between the species of the l-side and the species of the d-side. The existence of dual reactions is a consequence and is somewhat equivalent to the pseudochirality of network $\Omega_{C}$. Moreover, given a reaction $R_{L}$ and its dual $R_{D},$ the reaction rate constants of those two reactions are the same. In the particular case of Calvin’s mechanism, we have $k_{0}=k_{1}$, $k_{2}=k_{3}$, $k_{4}=k_{5}$, $k_{6}=k_{7}$, and $k_{8}=k_{9}$.

**Notation** **2.**
*In the Calvin model, we use the symbol $k_{0}$ to denote the reaction rate constant of the dual pair of autocatalytic reactions, the symbol $k_{2}$ to denote the reaction rate constant of their reverse reactions, the symbol $k_{4}$ to denote the reaction rate constant of the dual pair of racemization reactions, the symbols $k_{6}$ to denote the reaction rate constants of the enantiomeric conversions and $k_{8}$ to denote the reaction rate constants of their reverse reactions.*


We have that $\Omega_{C}$ is a pseudochiral network of order 2. Does network $\Omega_{C}$ exhibit homochiral dynamics? Calvin claimed that it is the case, but we think that the evidence provided by him is weak, and we would like to consider this question more carefully.

**Remark** **3.***Let* Ω *be a network, and suppose that* Ω *is not pseudochiral. Then, there is an asymmetry in* Ω *that distinguishes the L-side and the D-side. Thus, if* Ω *is not pseudochiral, it models a chemical mechanism that works on a chiral environment, and because of this it cannot be regarded as a network model of absolute asymmetric synthesis. We claim that: feasible network models of absolute asymmetric synthesis are pseudochiral networks.*

## 4. The Algorithmic Problem

Suppose that one wants to introduce a new network model of absolute asymmetric synthesis. One must show that this new model exhibits homochiral dynamics, and one must also show that the model is sound from a thermodynamical point of view. We focus on the first task, which is solved if one exhibits racemic steady states which, after being perturbed, trigger homochiral dynamics. Therefore, we consider the following algorithmic problem:

**Problem** **1.**
***CPVEH**: Computing the Parameter Values for the Emergence of Homochirality.*
*Input:* Ω*, where* Ω *is a pseudochiral network.**Problem: check if* Ω *has racemic steady states that give place to homochiral dynamics. In that case compute a sample of those states.*


It has been argued that the asymmetric synthesis of chiral biomolecules was a prerequisite for the origin of life [3]. It is supposed that there are chemical processes, which took place in prebiotic earth, and which transformed the initial racemic mixtures, present in prebiotic earth, into the homochiral mixtures that preceded the origin of life. Frank introduced in his seminal work [3] an abstract chemical mechanism, which contains an enantiomeric pair, and which evolves towards enantiopure states independently of the initial state. The chemical mechanism introduced by Frank is well described by the chemical reaction network $\Omega_{F}$. This network was the first network/mathematical model of absolute asymmetric synthesis introduced in the literature. After that, many other network models have been proposed, and we know that some of those proposed models are defective models that cannot support homochiral dynamics (we prove that the Calvin model [14] does not exhibit homochiral dynamics, see below). Therefore, we ask: can we recognize and discard all those defective models? To answer the question, we need an algorithm able to:Recognize the defective network models of absolute asymmetric synthesis that cannot exhibit homochiral dynamics.Recognize the network models that are mathematically sound, and compute samples of their sets of racemic steady states that undergo homochiral dynamics.

Thus, we need an algorithm that solves problem CPVEH.

## 5. The MM-Condition

Let Ω be a pseudochiral network, and suppose that we want to show that Ω has racemic steady states that produce homochiral dynamics. How can those states be found? Most authors cope with the latter question using the tools of classical stability analysis: the states that produce homochiral dynamics are *unstable* (see below).

**Remark** **4.***From now on we use the symbol Ω to denote a pseudochiral network of order k. Moreover, we suppose that*(18)$$\Omega =\left\{\left(L_{1},\ldots ,L_{k},D_{1},\ldots ,D_{k},X_{2k+1},\ldots ,X_{n}\right);\left(R_{1},\ldots ,R_{r}\right)\right\},$$*and we say that* Ω *is a network of size n.*

**Notation** **3.***We use the symbol $J_{\Omega }$ to denote the Jacobian of* Ω*. The Jacobian $J_{\Omega }$ is a symbolic matrix whose entries are the partial derivatives [8] of the reaction rates [15]. The entries of $J_{\Omega }$ are polynomials over the variables*
(19)$$\left[L_{1}\right],\ldots ,\left[L_{k}\right],\left[D_{1}\right],\ldots ,\left[D_{k}\right],\left[X_{2k+1}\right],\ldots ,\left[X_{n}\right],k_{1},\ldots ,k_{r}.$$*Thus, given a state s, one can evaluate $J_{\Omega }$ at s and obtain a numerical matrix $J_{\Omega }\left(s\right)$. We say that $J_{\Omega }\left(s\right)$ is the Jacobian of* Ω *at (state) s.*

Let us consider the pseudochiral network $\Omega_{C}$. The steady state equations for the l-species are equal to:(20)$$\begin{array}{ccc} 0 & = & \frac{d\left[L_{1}\right]}{dt}=-k_{0}\left[L_{1}\right]\left[L_{2}\right]+k_{2}{\left[L_{2}\right]}^{2}-k_{4}\left(\left[L_{1}\right]-\left[D_{1}\right]\right)-k_{6}\left[L_{1}\right]+k_{8}\left[L_{2}\right], \end{array}$$
(21)$$\begin{array}{ccc} 0 & = & \frac{d\left[L_{2}\right]}{dt}=k_{0}\left[L_{1}\right]\left[L_{2}\right]-k_{2}{\left[L_{2}\right]}^{2}+k_{6}\left[L_{1}\right]-k_{8}\left[L_{2}\right]. \end{array}$$

In addition, if we assume the racemic condition, those equalities get equal to
(22)$$0=-\frac{d\left[L_{1}\right]}{dt}=\frac{d\left[L_{2}\right]}{dt}=k_{0}\left[L_{1}\right]\left[L_{2}\right]-k_{2}{\left[L_{2}\right]}^{2}+k_{6}\left[L_{1}\right]-k_{8}\left[L_{2}\right],$$
while the Jacobian matrix gets equal to
(23)$$\left[\begin{array}{cccc} {-k}_{0}\left[L_{2}\right]{-k}_{4}-k6 & {-k}_{0}\left[L_{1}\right]{+2k}_{2}\left[L_{2}\right]{+k}_{8} & k_{4} & 0\\ k_{0}\left[L_{2}\right]{+k}_{6} & k_{0}\left[L_{1}\right]{-2k}_{2}\left[L_{2}\right]{-k}_{8} & 0 & 0\\ k_{4} & 0 & {-k}_{0}\left[L_{2}\right]{-k}_{4}-k6 & {-k}_{0}\left[L_{1}\right]{+2k}_{2}\left[L_{2}\right]{+k}_{8}\\ 0 & 0 & k_{0}\left[L_{2}\right]{+k}_{6} & k_{0}\left[L_{1}\right]{-2k}_{2}\left[L_{2}\right]{-k}_{8} \end{array}\right].$$

**Remark** **5.***The reader must observe the symmetrical structure of the above matrix, which we call the racemic Jacobian of $\Omega_{C}.$ It happens that the racemic Jacobian of any pseudochiral network* Ω *reveals the same type of symmetries [9].*

The Jacobian matrix $J_{\Omega }$ encodes important information related to the dynamics of Ω:

**Definition** **6.**
*We say that a steady state s is unstable, if and only if, the spectrum of $J_{\Omega }\left(s\right)$ satisfies the following:*

*There exists λ that belongs to the spectrum of $J_{\Omega }\left(s\right)$ and such that the inequality Reλ>0 holds.*

*We say that a polynomial $p\left(X\right)$ is unstable, if and only if, the roots of $p\left(X\right)$ satisfy the same condition imposed on the spectrum of the unstable states. Notice that s is unstable, if and only if, the characteristic polynomial of $J_{\Omega }\left(s\right)$ is unstable.*

*Recall that the spectrum of a square matrix is the set constituted by all its eigenvalues.*


The unstable states are the states that can get dramatically transformed by the effect of negligible perturbations. We use the term *symmetry-breaking states* to designate the racemic steady states that can produce homochiral dynamics. Notice that those homochiral dynamics dramatically transform those racemic states. Thus, we get that any symmetry-breaking state is unstable. However, we must observe that there exist unstable states that do not produce homochiral dynamics. We must ask: which are the symmetry-breaking states of network Ω? Which is the mathematical condition that determines the symmetry-breaking status? We observed before that the symmetry-breaking status depends on the eigenvectors of $J_{\Omega }\left(s\right)$ and not only on its eigenvalues, see [9,16,17].

**Definition** **7.**
*We say that a steady state s is symmetry-breaking, if and only if, matrix $J_{\Omega }\left(s\right)$ satisfies:*
*1.* 
*There exists λ that belongs to the spectrum of $J_{\Omega }\left(s\right)$, and such that Reλ>0.*
*2.* 
*There exists an eigenvector of λ, say v, and there exists i≤k such that $v\left[i\right]\neq v\left[i+k\right]$.*



We get that the symmetry-breaking states of Ω are the unstable states that satisfy a further constraint. The additional constraint refers to the eigenvectors of $J_{\Omega }\left(s\right)$. In principle, this additional constraint should make harder the search of those states. However, and surprisingly, it is not the case: if one exploits the symmetries of $J_{\Omega }$, the latter problem becomes easier than the former.

**Notation** **4.***Let* Ω *be a pseudochiral network of order k (recall that the order k is equal to the number of enantiomeric pairs, k=2 for the Calvin model), we use the symbol $A_{\Omega }$ to denote the submatrix of $J_{\Omega }$ that is constituted by its first k rows and its first k columns. We use the symbol $B_{\Omega }$ to denote the submatrix that is constituted by the first k rows of $J_{\Omega }\left(s\right)$ and the columns k+1,…,2k. For instance, if we consider Calvin network $\Omega_{C}$, we get that*
(24)$$\begin{array}{ccc} A_{\Omega_{C}} & = & \left[\begin{array}{cc} -k_{0}\left[L_{2}\right]-k_{4}-k_{6} & -k_{0}\left[L_{1}\right]+2k_{2}\left[L_{2}\right]+k_{8}\\ k_{0}\left[L_{2}\right]+k_{6} & k_{0}\left[L_{1}\right]-2k_{2}\left[L_{2}\right]-k_{8} \end{array}\right]\;and \end{array}$$
(25)$$\begin{array}{ccc} B_{\Omega_{C}} & = & \left[\begin{array}{cc} k_{4} & 0\\ 0 & 0 \end{array}\right]. \end{array}$$

**Theorem** **1.***(**MM-condition**) Let s be a racemic steady state of the pseudochiral network* Ω*, we have that s is symmetry-breaking, if and only if, the characteristic polynomial of $A_{\Omega }\left(s\right)-B_{\Omega }\left(s\right)$ is unstable [9].*

The above theorem allows us to analyze any pseudochiral network. Let us illustrate the latter claim with the analysis of the network $\Omega_{C}$.

The symmetry-breaking states of $\Omega_{C}$ are the 9-tuples
(26)$$\left(\left[L_{1}\right],\left[L_{2}\right],\left[D_{1}\right],\left[D_{2}\right],k_{0},k_{2},k_{4},k_{6},k_{8}\right)$$
that satisfy the following constraints:The racemic condition
(27)$$\left[L_{1}\right]=\left[D_{1}\right]\;\mathrm{and}\;\left[L_{2}\right]=\left[D_{2}\right].$$The steady state condition
(28)$$k_{0}\left[L_{1}\right]\left[L_{2}\right]+k_{6}\left[L_{1}\right]=k_{2}{\left[L_{2}\right]}^{2}+k_{8}\left[L_{2}\right].$$The *non-negativity condition*
(29)$$\left[L_{1}\right],\left[L_{2}\right],\left[D_{1}\right],\left[D_{2}\right]\geq 0.$$The *positivity condition*
(30)$$k_{0},\;k_{2},\;k_{4},\;k_{6},\;k_{8}>0$$The (*symmetry-breaking*) *MM-condition* asserting that the characteristic polynomial of matrix
(31)$$A_{\Omega_{C}}-B_{\Omega_{C}}=\left[\begin{array}{cc} -k_{0}\left[L_{2}\right]-2k_{4}-k_{6} & -k_{0}\left[L_{1}\right]+2k_{2}\left[L_{2}\right]+k_{8}\\ k_{0}\left[L_{2}\right]+k_{6} & k_{0}\left[L_{1}\right]-2k_{2}\left[L_{2}\right]-k_{8} \end{array}\right]$$
is unstable.

Notice that the first four constraints are polynomial inequalities (equalities) over the variables $\left[L_{1}\right],\left[L_{2}\right],\left[D_{1}\right],\left[D_{2}\right],\;k_{0},\;k_{2},\;k_{4},\;k_{6},\;\;\mathrm{and}\;k_{8}$. The last constraint can also be expressed in terms of polynomial inequalities. To this end one can use a suitable set of *Hurwitz-Routh inequalities* [18]. We have, for instance, that the 2×2 matrix $A_{\Omega_{C}}-B_{\Omega_{C}}$ is unstable, if and only if, at least one of the following two inequalities is satisfied:$k_{0}\left[L_{2}\right]+2k_{4}+k_{6}-k_{0}\left[L_{1}\right]+2k_{2}\left[L_{2}\right]+k_{8}<0$$k_{8}-k_{0}\left[L_{1}\right]+2k_{2}\left[L_{2}\right]<0$ .

It is easy to prove that there are no steady states satisfying at least one of the above inequalities.

**Theorem** **2.**
*The Calvin model does not have symmetry-breaking states.*


**Proof** **of** **Theorem** **19.**First we consider the inequality
(32)$$k_{8}-k_{0}\left[L_{1}\right]+2k_{2}\left[L_{2}\right]<0.$$Let us suppose that $\left[L_{2}\right]\neq 0$. From the steady state condition, Equation (Equation 28), we get that
(33)$$k_{0}\left[L_{1}\right]=k_{8}+k_{2}\left[L_{2}\right]-k_{6}\frac{\left[L_{1}\right]}{\left[L_{2}\right]}<k_{8}+2k_{2}\left[L_{2}\right],$$
and hence the Inequality (Equation 32) cannot be satisfied.On the other hand, if we suppose $\left[L_{2}\right]=0,$ we get that the symmetry-breaking states of $\Omega_{C}$ must be solutions of the system
(34)$$k_{6}\left[L_{1}\right]=0\;\mathrm{and}\;k_{0}\left[L_{1}\right]>k_{8},$$
which does not have solutions satisfying the constraints $k_{0},k_{6},k_{8}>0$ and $\left[L_{1}\right]\geq 0$.Now, we consider the inequality
(35)$$k_{0}\left[L_{2}\right]+2k_{4}+k_{6}-k_{0}\left[L_{1}\right]+2k_{2}\left[L_{2}\right]+k_{8}<0.$$We get from the steady state condition (Equation 28) that
(36)$$k_{0}\left[L_{1}\right]\leq k_{2}\left[L_{2}\right]+k_{8},$$
and we get consequently that
(37)$$k_{0}\left[L_{1}\right]<k_{2}\left[L_{2}\right]+k_{8}+2k_{4}+k_{6}+k_{0}\left[L_{2}\right].$$Then, the Inequality (Equation 32) cannot be satisfied, and the theorem is proved.  □

The analysis of the Calvin network, as developed in the previous paragraphs, shows that it is possible to use the MM-condition to analyze small networks thoroughly. What can be done with larger networks? The MM-condition yields an algorithmic solution for the CPVEH problem. However, if the input network Ω (mechanism) is too large, its analysis could be intractable. Consider the problem:

**Problem** **2.**
***SA: Stability Analysis***

*Input: M, where M is a symbolic matrix such that all its entries are polynomials.*

*Problem: Determine whether the set of parameter values, which make the characteristic polynomial of M unstable, is nonempty. If this is the case, sample this set.*



The MM-condition allows us to reduce the problem CPVEH to this latter problem. However, this reduction can be useless, given that problem SA is a hard, intractable problem. Thus, we must ask: can we efficiently solve problem CPVEH?

We claimed before that computing the symmetry-breaking states of Ω can be easier than computing its unstable states. The MM-condition strongly reduces the dimensionality of the problem. This mathematical criterion tells us that we must analyze a k×k symbolic matrix instead of the n×n symbolic matrix that we would have to analyze if we were to use classical stability analysis to compute, as in previous approaches, the unstable states of Ω (notice that n≥2k). It seems that it is not possible to further reduce the dimensionality of the problem, and it forces us to consider other completely different reductions. We must observe, at this point, that there are two main sources of complexity related to the instances of problem SA: the dimension of the input matrices, and the degree of their polynomial entries. Thus, we ask: can we also reduce the inherent nonlinearity of the instances of CPVEH? In the following section we introduce some tools of Clarke’s *Stoichiometric Network Analysis* (SNA, see reference [10]), which allow us to achieve the degree reduction we are looking for. We use SNA, and the degree reduction provided by it, to develop an algorithm that can be used in the analysis of pseudochiral networks.

## 6. Degree Reduction Using Stoichiometric Network Analysis

Clarke’s SNA provides us with tools that can be used in the linear stability analysis of chiral networks, see [19] and the references therein. Here, we use SNA to develop a heuristic algorithm for the CPVEH problem.

### A Crash Introduction to SNA

Let Θ be a chemical reaction network and let $\mathrm{SS}\left(\Theta \right)$ be its set of steady states. SNA is based on a change of variables that linearize the definition of $\mathrm{SS}\left(\Theta \right)$: this change of variables maps the latter set onto a *polyhedral cone* that we denote with the symbol $C_{\Theta }.$

**Notation** **5.***Along this section we use the symbol* Θ *to denote a chemical reaction network. Moreover, we suppose that*
(38)$$\Theta =\left\{\left(X_{1},\ldots ,X_{n}\right);\left(R_{1},\ldots ,R_{r}\right)\right\},$$
*and we suppose that $R_{i}$ is equal to*
(39)$$c_{1i}X_{{}_{1}}+\cdots +c_{ni}X_{{}_{n}}\to d_{1i}X_{{}_{1}}+\cdots +d_{ni}X_{n}.$$

**Definition** **8.***The stoichiometric matrix of* Θ *that we denote with the symbol $S_{\Theta }$, is a n×r matrix*
(40)$$S_{\Theta }={\left[\nu_{ij}\right]}_{i\leq n,j\leq r}\;,$$
*whose entries are called the stoichiometric coefficients of* Θ*. The stoichiometric coefficients of the network* Θ *are defined by:*
(41)$$\nu_{ij}=d_{ij}-c_{ij}.$$
*A second matrix related to $S_{\Theta }$ is the matrix of orders of reaction denoted with the symbol $R_{\Theta }$, and which is the r×n matrix*
(42)$$R_{\Theta }={\left[c_{ji}\right]}_{i\leq n,j\leq r}.$$


**Definition** **9.***The velocity function of* Θ *is the function $V_{\Theta }:R^{n}\to R^{s}$ that is defined by*
(43)$$V_{\Theta }\left(\left[\vec{X}\right]\right)=\left(k_{1}{\left[X_{1}\right]}^{c_{11}}\cdots {\left[X_{n}\right]}^{c_{n1}},\ldots ,k_{r}{\left[X_{1}\right]}^{c_{1r}}\cdots {\left[X_{n}\right]}^{c_{nr}}\right).$$

Clarke observed that the dynamic equation of Θ can be written as
(44)$$\frac{d\left[\vec{X}\right]}{dt}=S_{\Theta }\cdot V_{\Theta }\left(\left[\vec{X}\right]\right),$$
and it implies that the function $V_{\Theta }$ maps the set $\mathrm{SS}\left(\Theta \right)$ onto the *polyhedral cone*
$C_{\Theta }$ that is defined by the linear constraints
(45)$$\begin{array}{ccc} V_{1},\ldots ,V_{r} & \geq & 0,\;\mathrm{and} \end{array}$$
(46)$$\begin{array}{ccc} S_{\Theta }\cdot \vec{V} & = & 0. \end{array}$$

Thus, the function $V_{\Theta }$ allows us to identify the steady states of Θ with the points of $C_{\Theta }$, and it happens that the points of $C_{\Theta }$ can be suitably described by the system of *convex coordinates* provided by its *extreme currents*[10].

**Remark** **6.**
*A set of extreme currents for $C_{\Theta }$ is just a minimal set of extreme rays that spans the polyhedral cone $C_{\Theta }$, see [20].*


**Notation** **6.**
*We use the symbol $R_{+}^{s}$ to denote the set of s-dimensional nonnegative vectors, which is the set*
(47)$$\left\{v\in R^{s}:v\left[1\right],\ldots ,v\left[s\right]\geq 0\right\}.$$


**Definition** **10.**
*Let $v_{1},\ldots ,v_{s}$ be a set of extreme currents for $C_{\Theta }$, and let $s\in \mathrm{SS}\left(\Theta \right)$; the convex coordinates of s are given by the unique tuple $\left(j_{1},\ldots ,j_{s}\right)\in R_{+}^{s}$ that satisfies the equality*
(48)$$V_{\Theta }\left(s\right)=j_{1}v_{1}+\cdots +j_{s}v_{s}.$$

*From now on, we use the symbols $j_{1},\ldots ,j_{s}$ to denote the convex coordinates of the cone $C_{\Theta }$ that are determined by $v_{1},\ldots ,v_{s}.$*


We have:Given $s\in \mathrm{SS}\left(\Theta \right),$ one can effectively compute the convex coordinates of s.Given $\left(j_{1},\ldots ,j_{s}\right)\in R_{+}^{s},$ one can effectively sample the set $V_{\Theta }^{-1}\left(j_{1}v_{1}+\cdots +j_{s}v_{s}\right)$, which is a nonempty subset of $\mathrm{SS}\left(\Theta \right)$.

If one switches to the system of convex coordinates, then the Jacobian $J_{\Theta }$ can be factorized as
(49)$$J_{\Theta }=S_{\Theta }\cdot E_{\Theta }\cdot R_{\Theta }\cdot \Delta ,$$
where Δ is a n×n scaling matrix, and $E_{\Theta }$ is the r×r diagonal matrix defined by
(50)$$E_{\Theta }\left[i,i\right]=\sum_{l\leq s}v_{l}\left[i\right]\cdot j_{l}.$$

**Remark** **7.**
*The term $E_{\Theta }\left[i,i\right]$ is a linear polynomial over the (convex) variables $j_{1},\ldots ,j_{s}.$*


Clarke noticed that the scaling matrix Δ has little influence on the stability properties of $J_{\Theta }$, and that one can focus the analysis on the matrix $S_{\Theta }\cdot E_{\Theta }\cdot R_{\Theta }$. Thus, according to Clarke’s theory, the stability analysis of Θ reduces to the stability analysis of the latter matrix. We must ask: what do we gain with this reduction? The entries of $S_{\Theta }\cdot E_{\Theta }\cdot R_{\Theta }$ are linear polynomials over the variables $j_{1},\ldots ,j_{s}$, and it means that we get an important degree reduction.

We use the symbol $V_{\Theta }$ to denote the matrix $S_{\Theta }\cdot E_{\Theta }\cdot R_{\Theta }.$ The SNA-based algorithmic analysis of chemical networks reduces to:Compute a set of extreme currents for $C_{\Theta }.$Compute the symbolic matrix $V_{\Theta }.$Sample the set of parameter values that make the characteristic polynomial of matrix $V_{\Theta }$ become unstable.Given α, a *s*-dimensional vector that belongs to the sample computed in step 3, compute a sample of the set $V_{\Theta }^{-1}\left(\alpha_{1}v_{1}+\cdots +\alpha_{s}v_{s}\right).$ Here, we use the symbols $v_{1},\ldots ,v_{s}$ to denote the set of extreme currents for $C_{\Theta }$ as computed in step 1.

**Remark** **8.**
*The latter algorithmic routine can be effectively implemented, see [11].*


#### An Illustrative and Trivial Example

Let $\Theta_{0}$ be the network $\left\{\left(I,A\right),\left(R_{1},R_{2},R_{3}\right)\right\}$, where:

reaction $R_{1}$:3I →[k1] 3Areaction $R_{2}$:2I + A →[k2] 3I, andreaction $R_{3}$:I + A →[k3] 2A.

Clarke’s velocity function denoted by $V_{\Theta_{0}}$ is equal to the vector field $\left(k_{1}{\left[I\right]}^{3},k_{2}{\left[I\right]}^{2}\left[A\right],k_{3}\left[I\right]\left[A\right]\right)$. The set $\mathrm{SS}\left(\Theta_{0}\right)$ is the subset of $R_{+}^{2}\times R_{+}^{3}$ that is defined by the equation
(51)$$-3k_{1}{\left[I\right]}^{3}+k_{2}{\left[I\right]}^{2}\left[A\right]-k_{3}\left[I\right]\left[A\right]=0.$$

**Remark** **9.**
*Notice that $\mathrm{SS}\left(\Theta_{0}\right)$ is a 5-dimensional set with a quite complex structure.*


The stoichiometric matrix of $\Theta_{0}$ is the matrix
(52)$$S_{\Theta_{0}}=\begin{array}{cc} & \begin{array}{ccc} R_{1} & R_{2} & R_{3} \end{array}\\ \begin{array}{c} \left[I\right]\\ \left[A\right] \end{array} & \left[\begin{array}{ccc} -3 & 1 & -1\\ 3 & -1 & 1 \end{array}\right] \end{array}$$
and a set of extreme currents for $C_{\Theta_{0}}$ is given by $\left\{\left(1,3,0\right),\left(0,1,1\right)\right\}.$ This means that the 5-dimensional set $\mathrm{SS}\left(\Theta_{0}\right)$ is mapped by $R_{\Theta_{0}}$ onto the infinite triangle spanned by the vectors $\left(1,3,0\right)$ and $\left(0,1,1\right).$ Thus, we get that $V_{\Theta_{0}}\left(\mathrm{SS}\left(\Theta_{0}\right)\right)$ has a pleasant conic structure. We can use the extreme currents of $C_{\Theta_{0}}$ (which are integer vectors located on the borders of $C_{\Theta_{0}},$ for instance the vectors $\left(1,3,0\right)$ and $\left(0,1,1\right)$) to define a system of convex coordinates for $C_{\Theta_{0}}.$ Notice that the points of this triangle are in bijective correspondence with the nonnegative linear combinations of the extreme currents $\left(1,3,0\right)$ and $\left(0,1,1\right)$. Thus, if we choose nonnegative values for $j_{1},j_{2}$, we can be sure that
(53)$$j_{1}\cdot \left(1,3,0\right)+j_{2}\cdot \left(0,1,1\right)$$
is an element of $C_{\Theta_{0}}$ that represents a nonempty set of steady states. We can use this fact to sample the set $\mathrm{SS}\left(\Theta_{0}\right)$. We proceed in the following way: *Pick a tuple of positive convex coordinates; for instance*, $j_{1}=1$*and*$j_{2}=2$.*Compute the corresponding convex combination of extreme rays; in our case, we compute*$1\cdot \left(1,3,0\right)+2\left(0,1,1\right)$*equal to*$\left(1,5,2\right)$.*Compute the solution set of the nonlinear system*(54)$$\begin{array}{ccc} 1 & = & k_{1}{\left[I\right]}^{3} \end{array}$$(55)$$\begin{array}{ccc} 5 & = & k_{2}{\left[I\right]}^{2}\left[A\right] \end{array}$$(56)$$\begin{array}{ccc} 2 & = & k_{3}\left[I\right]\left[A\right]. \end{array}$$*Pick an element, say*$\left(1,1,1,5,2\right)$, *of the set computed in the previous step*.

This means that given a chemical network Θ, we can use Clarke’s velocity function to define a system of convex coordinates for the set $\mathrm{SS}\left(\Theta \right)$. Moreover, we can use this system of coordinates to sample the set $\mathrm{SS}\left(\Theta \right)$. We can also use the factorization of $J_{\Theta }$ to analyze the stability of network Θ.

Additionally, it is important to illustrate how this procedure reduces the degree of the polynomials entries of the symbolic Jacobian $J_{\Theta_{0}}$. In our illustrative and trivial example, the symbolic Jacobian is equal to
(57)$$J_{\Theta_{0}}=\left[\begin{array}{cc} -9k_{1}{\left[I\right]}^{2}+2k_{2}\left[I\right]\left[A\right]-k_{3}\left[A\right] & k_{2}{\left[I\right]}^{2}-k_{3}\left[I\right]\\ 9k_{1}{\left[I\right]}^{2}-2k_{2}\left[I\right]\left[A\right]+k_{3}\left[A\right] & -k_{2}{\left[I\right]}^{2}+k_{3}\left[I\right] \end{array}\right].$$

We have that $R_{\Theta_{0}}$ is equal to
(58)$$R_{\Theta_{0}}=\begin{array}{cc} & \begin{array}{cc} \left[I\right] & \left[A\right] \end{array}\\ \begin{array}{c} R_{1}\\ R_{2}\\ R_{3} \end{array} & \left[\begin{array}{cc} 3 & 0\\ 2 & 1\\ 1 & 1 \end{array}\right] \end{array},$$
and we have that $E_{\Theta_{0}}$ is equal to
(59)$$E_{\Theta_{0}}=\left[\begin{array}{ccc} j_{1} & 0 & 0\\ 0 & 3j_{1}+j_{2} & 0\\ 0 & 0 & j_{2} \end{array}\right].$$

According to Clarke’s factorization the matrix $J_{\Theta_{0}}$ is equal to
(60)$$J_{\Theta_{0}}=\left[\begin{array}{ccc} -3 & 1 & -1\\ 3 & -1 & 1 \end{array}\right]\cdot \left[\begin{array}{ccc} j_{1} & 0 & 0\\ 0 & 3j_{1}+j_{2} & 0\\ 0 & 0 & j_{2} \end{array}\right]\cdot \left[\begin{array}{cc} 3 & 0\\ 2 & 1\\ 1 & 1 \end{array}\right]\cdot \left[\begin{array}{cc} \frac{1}{i} & 0\\ 0 & \frac{1}{a} \end{array}\right],$$
and this factorization allows us to focus the stability analysis on the matrix
(61)$$V_{\Theta_{0}}=\left[\begin{array}{ccc} -3 & 1 & -1\\ 3 & -1 & 1 \end{array}\right]\cdot \left[\begin{array}{ccc} j_{1} & 0 & 0\\ 0 & 3j_{1}+j_{2} & 0\\ 0 & 0 & j_{2} \end{array}\right]\cdot \left[\begin{array}{cc} 3 & 0\\ 2 & 1\\ 1 & 1 \end{array}\right],$$
which is equal to
(62)$$V_{\Theta_{0}}=\left[\begin{array}{cc} -3j_{1}+j_{2} & 3j_{1}\\ 3j_{1}-j_{2} & -3j_{1} \end{array}\right]$$

Notice that the entries of the matrix $V_{\Theta_{0}}$ are linear polynomials over the variables $j_{1},j_{2},$ while the entries of $J_{\Theta_{0}}$ are 3-degree polynomials over the parameters $\left[I\right],\left[A\right],k_{1},k_{2}$ and $k_{3}$. Thus, it is true that we gain an important degree reduction if we switch to the system of convex coordinates.

## 7. Using SNA in the Analysis of Pseudochiral Networks

We use the tools introduced in the previous paragraphs to develop an algorithm for the stability (symmetry-breaking) analysis of pseudochiral networks.

Let us consider the case of pseudochiral networks of order 1, which we call chiral networks. Thus, let Φ be a chiral network of order 1, and suppose that
(63)$$\Phi =\left\{\left(L_{1},D_{1},X_{3},\ldots ,X_{n}\right);\left(R_{1},\ldots ,R_{r}\right)\right\}.$$

The MM-condition reduces to the inequality
(64)$$J_{\Phi }\left[1,1\right]-J_{\Phi }\left[1,2\right]>0\;,$$
which is a linear inequality over the entries of $J_{\Phi }.$ However, the linearity of this algebraic condition is just apparent, given that the terms $J_{\Phi }\left[1,1\right]$ and $J_{\Phi }\left[1,2\right]$ are nonlinear polynomials over the concentration variables
(65)$$\left[L_{1}\right],\left[D_{1}\right],\left[X_{3}\right],\ldots ,\left[X_{n-1}\right]and\left[X_{n}\right].$$

**Notation** **7.**
*We use the symbol $\mathrm{SB}\left(\Phi \right)$ to denote the set of symmetry-breaking states of Φ.*


We have that the set $\mathrm{SB}\left(\Phi \right)$ is a *highly nonlinear set* defined by the conditions:The nonlinear steady state equations.The nonlinear inequality
(66)$$J_{\Phi }\left[1,1\right]-J_{\Phi }\left[1,2\right]>0\;.$$The non-negativity conditions
(67)$$\left[L_{1}\right],\left[D_{1}\right],\left[X_{3}\right],\ldots ,\left[X_{n}\right]\geq 0\;.$$The positivity conditions
(68)$$k_{1},\ldots ,k_{r}>0\;.$$The racemic condition stating that the rate constants of the reactions that belong to the same dual pair are equal, and that the initial concentrations of the two enantiomeric species are also equal.

However, if we switch to Clarke’s system of convex coordinates, we get that this set is defined by the linear conditions:$j_{1},\ldots ,j_{s}\geq 0,$ where $j_{1},\ldots ,j_{s}$ constitute the system of convex coordinates of network Φ.$V_{\Phi }\left[1,1\right]-V_{\Phi }\left[1,2\right]>0.$

**Remark** **10.**
*In the implementation of the algorithm, we forced condition five (the racemic condition) into the computation of the extreme currents, which are computed from the stoichiometric matrix; for this, we extended the stoichiometric matrix with rows that encode the equalities of the rate constants of the dual pairs; for example, given the stoichiometric matrix of the Calvin model*
(69)$$S_{\Omega_{C}}=\begin{array}{cc} & \begin{array}{cccccccccc} R_{0} & R_{1} & R_{2} & R_{3} & R_{4} & R_{5} & R_{6} & R_{7} & R_{8} & R_{9} \end{array}\\ \begin{array}{c} L_{1}\\ D_{1}\\ L_{2}\\ D_{2} \end{array} & \left[\begin{array}{cccccccccc} -1 & 0 & 1 & 0 & -1 & 1 & -1 & 0 & 1 & 0\\ 0 & -1 & 0 & 1 & 1 & -1 & 0 & -1 & 0 & 1\\ 1 & 0 & -1 & 0 & 0 & 0 & 1 & 0 & -1 & 0\\ 0 & 1 & 0 & -1 & 0 & 0 & 0 & 1 & 0 & -1 \end{array}\right] \end{array}\;,$$
*the extended matrix, considering its dual pairs (Reactions (Equation 13), (Equation 14), (Equation 16), (Equation 17)), is equal to*
(70)$$S_{\Omega_{C}}=\begin{array}{cc} & \begin{array}{cccccccccc} R_{0} & R_{1} & R_{2} & R_{3} & R_{4} & R_{5} & R_{6} & R_{7} & R_{8} & R_{9} \end{array}\\ \begin{array}{c} L_{1}\\ D_{1}\\ L_{2}\\ D_{2}\\ R_{0}\;and\;R_{1}\\ R_{2}\;and\;R_{3}\\ R_{6}\;and\;R_{7}\\ R_{8}\;and\;R_{9} \end{array} & \left[\begin{array}{cccccccccc} -1 & 0 & 1 & 0 & -1 & 1 & -1 & 0 & 1 & 0\\ 0 & -1 & 0 & 1 & 1 & -1 & 0 & -1 & 0 & 1\\ 1 & 0 & -1 & 0 & 0 & 0 & 1 & 0 & -1 & 0\\ 0 & 1 & 0 & -1 & 0 & 0 & 0 & 1 & 0 & -1\\ 1 & -1 & 0 & 0 & 0 & 0 & 0 & 0 & 0 & 0\\ 0 & 0 & 1 & -1 & 0 & 0 & 0 & 0 & 0 & 0\\ 0 & 0 & 0 & 0 & 0 & 0 & 1 & -1 & 0 & 0\\ 0 & 0 & 0 & 0 & 0 & 0 & 0 & 0 & 1 & -1 \end{array}\right] \end{array}\;.$$

*Forcing this condition into the computations of the extreme currents proved to reduce the number of extreme currents significantly.*

*The stoichiometric and extended stoichiometric matrices of the models analyzed with Listanalchem can be seen in the output of the computer program. The Appendix A presents those matrices for the models studied in this work.*


Sets defined by linear inequalities are easy to sample, see [21]. Then, if we switch to the system of convex coordinates, we will be able to efficiently sample the set of symmetry-breaking states of Φ. Thus, we can use the MM-condition and SNA to efficiently solve the CPVEH problem when it is restricted to pseudochiral networks of order 1.

What about higher orders? Suppose that Ω is a pseudochiral network of order *k*, suppose that the variables $j_{1},\ldots ,j_{s}$ constitute a system of convex coordinates for $C_{\Omega }$ and let
(71)$$AV_{\Omega }={\left[V_{\Omega }\left[i,j\right]\right]}_{i,j\leq k}\;,\;\;\mathrm{and}\;\;BV_{\Omega }={\left[V_{\Omega }\left[i,j\right]\right]}_{i\leq k;k+1\leq j\leq 2k}$$
be the appropriate submatrices of $V_{\Omega }$. The set $\mathrm{SB}\left(\Omega \right)$ is defined by the algebraic conditions:$j_{1},\ldots ,j_{s}\geq 0$.The characteristic polynomial of the k×k matrix $AV_{\Omega }-BV_{\Omega }$ is unstable.

Notice that we get a strong reduction on the degree of the polynomial expressions defining the set of symmetry-breaking states. The latter is the case given that:We eliminated all the nonlinearity that was implicit in the definition of the set of steady states.We reduced the degree of the instability condition. The instability condition is given by the *Hurwitz-Routh inequalities* [18], and those inequalities involve the determinant of the matrix to be analyzed. The entries of $J_{\Omega }$ are nonlinear polynomials, and the determinant of this symbolic matrix is a polynomial whose degree can be larger than 2k. On the other hand, we have that the entries of $AV_{\Omega }-BV_{\Omega }$ are linear polynomials, and we have that the degree of $\det\left(AV_{\Omega }-BV_{\Omega }\right)$ is upperbounded by k.

However, if *k* is large, and despite all the reductions achieved so far, the stability analysis of matrix $AV_{\Omega }-BV_{\Omega }$ can be unfeasible. A full stability analysis of matrix $AV_{\Omega }-BV_{\Omega }$ presupposes the computation of its determinant, as well as the computation of other large subdeterminants. Take into account that computing symbolic determinants requires exponential time. If we want to avoid the computation of those large (symbolic) determinants, we will have to conform ourselves with a heuristic algorithm.

### A Heuristic Algorithm for CPVEH Based on SNA and the MM-Condition

Let Ω be a pseudochiral network of order *k*, let $AV_{\Omega }-BV_{\Omega }$ be the symbolic matrix that we want to analyze and let
(72)$$p_{\Omega }\left(\lambda \right)=\lambda^{k}-\Omega_{1}\lambda^{k-1}+\Omega_{2}\lambda^{k-2}+\cdots +{\left(-1\right)}^{k}\Omega_{k}\lambda^{k-k}$$
be the characteristic polynomial of this matrix.

The coefficient $\Omega_{i}$ is equal to the sum of all the diagonal subdeterminants of $AV_{\Omega }-BV_{\Omega }$ of order *i*, see [22]. Notice that all those coefficients are polynomials over the set of parameters that we are analyzing, and notice that we are interested in determining the values of those parameters that make the latter (parameterized) polynomial become unstable. We must take into account the following fact: the roots of a polynomial are determined by its coefficients. Let us consider two instances of the latter phenomenon:If the inequality ${\left(-1\right)}^{k}\Omega_{k}<0$ holds, the polynomial $p_{\Omega }\left(\lambda \right)$ has a positive real root.$p_{\Omega }\left(\lambda \right)$ can have positive real roots only if it has negative coefficients.

We could focus on the first item and conform ourselves with a sufficient condition for instability, or we could focus on the second item and conform ourselves with a necessary condition for the existence of positive real roots. Notice that $\Omega_{k}$ is equal to $\det\left(AV_{\Omega }-BV_{\Omega }\right)$ and recall that we wanted to avoid the computation of large subdeterminants of the matrix $AV_{\Omega }-BV_{\Omega }$. This latter observation leads us to focus on the second item and consider the set ${\mathrm{SB}}^{*}\left(\Omega \right)$, which is the subset of $\mathrm{SB}\left(\Omega \right)$ constituted by the steady states that satisfy the condition: there exists i≤k such that ${\left(-1\right)}^{i}\Omega_{i}<0$. We must ask: what do we gain if we focus on the second criterion? It was observed that in most cases, the chemical instabilities of Ω are determined by small subnetworks, see [23]. Moreover, given i≤k, the influence of all the subnetworks of size *i* is encoded in the coefficient $\Omega_{i}$. We can conclude that in most cases, the chemical instabilities of Ω are encoded in the first coefficients of $p_{\Omega }\left(\lambda \right)$. We can use the latter as a further heuristic principle which tells us that: for most pseudochiral networks exhibiting homochiral dynamics, there exist small values of *i* such that the inequality ${\left(-1\right)}^{i}\Omega_{i}<0$ gets satisfied for nonnegative values of the parameters. Therefore, we focus on the problem:

**Problem** **3.**
***approx-CPVEH***
*Input:* Ω*, where* Ω *is a pseudochiral network.*
*Problem: compute the minimum i for which the inequality ${\left(-1\right)}^{i}\Omega_{i}<0$ gets satisfied for nonnegative values of the parameters, and sample this set of nonnegative solutions.*



Our heuristic approach for solving CPVEH consists of solving the problem approx-CPVEH. Consider the following algorithm that we denote with the symbol SNA-sampling.

*Algorithm SNA-sampling works, on input*Ω, *as follows:**Compute the matrices*$AV_{\Omega }$*and*$BV_{\Omega }$.*Given*i≤5*determine the set of nonnegative solutions of the inequality*${\left(-1\right)}^{i}\Omega_{i}<0$*. If all those sets are empty reject*Ω; *otherwise, sample the union of those five sets*.*Given*$j=\left(j_{1},\ldots ,j_{s}\right)$*an element of the computed sample, compute a racemic steady state of*Ω,*say*$s_{j}$, *such that*j*is the tuple of convex coordinates of*$s_{j}$. *Do the same with all the members of the computed sample*.

We notice that:If Ω is a pseudochiral network of order 1, the algorithm correctly and efficiently samples the set $\mathrm{SB}\left(\Omega \right)$.If Ω is a pseudochiral network of order k≤5, then the algorithm samples the set ${\mathrm{SB}}^{*}\left(\Omega \right).$If Ω is a pseudochiral network of order k>5, then the algorithm approximates the set ${\mathrm{SB}}^{*}\left(\Omega \right)$ by a semialgebraic set that is defined by polynomial expressions whose degree is upperbounded by 5.

SNA-sampling is a heuristic algorithm that is as efficient as possible, solves the problem CPVEH for pseudochiral networks of small order, and allows us to compute important information in the case of higher orders. We exemplify, in the next section, the power of this tool with the analysis of three pseudochiral networks that were taken from the literature dedicated to the study of biological homochirality.

## 8. Computer Experiments

The algorithm SNA-sampling, as described above, was implemented as a computer program called Listanalchem [11], option six. The algorithm determines if a chemical network Ω, given as input, can produce spontaneous mirror-symmetric breaking (SMSB), and if so, samples the set of parameter values that can produce those dynamics. The computed samples are used to make numerical simulations of the dynamics. For this task, we use Chemkinlator [24], a software tool that can also build bifurcation diagrams. Actually, we used bifurcation diagrams to find the SMSB region for a model which presented problems for the heuristic used in sampling. The construction of the bifurcation diagrams could be a necessary additional step, given that, depending on the model, the instability region can be so small that numerical error could place us out of, but close, to this region. Also, the heuristic used in the algorithm can cause the results to be out of the instability region. In those cases, a fast exploration around the computed values, using bifurcation diagrams, could be enough to find SMSB. Additionally, this procedure can be used to build phase diagrams such as the ones shown in references [25,26].

It is worth remembering that as mentioned in Remark 10, the implementation of the algorithm uses the extended stoichiometric matrix with rows that encode the duality of the pairs of reactions that involve enantiomers. We observed that after adding those constraints the number of extreme currents got reduced. The details of this process, including the extended stoichiometric matrix, and its manipulation, can be seen in the output of Listanalchem option six. These outputs are available in the Appendix A.

Finally, we would like to remark that the units used for simulations are arbitrary. We do not set units to avoid huge or tiny numbers. Instead of that, we use numbers in the interval [0, 2]. This particular way to perform the analysis, for each model, does not change the qualitative behavior of the models, and it helps to show clearer images of the relevant facts. Particular units can be obtained using the corresponding factors in concentrations and rate constants, but that fact will be not explored here because we are interested, first of all, in the stability of the models that generate SMSB.

### 8.1. The Replicator Model of Hochberg and Ribo

Hochberg and Ribo [27] have investigated the network described below.

A + ^1^RD + ^2^RD $\overset{k_{0}}{\underset{k_{1}}{⇌}}$ 2^1^R_D_ + ^2^R_D_A + ^1^R_L_ + ^2^R_L_
$\overset{k_{2}}{\underset{k_{3}}{⇌}}$ 2^1^R_L_ + ^2^R_L_A + ^2^R_D_ + ^1^R_D_
$\overset{k_{4}}{\underset{k_{5}}{⇌}}$ 2^2^R_D_ + ^1^R_D_A + ^2^R_L_ + ^1^R_L_
$\overset{k_{6}}{\underset{k_{7}}{⇌}}$ 2^2^R_L_ +^ 1^R_L_^1^R_D_$\underset{\to }{k_{8}}$ ⌀^1^R_L_$\underset{\to }{k_{10}}$ ⌀^2^R_D_$\underset{\to }{k_{9}}$ ⌀^2^R_L_$\underset{\to }{k_{11}}$ ⌀$\bar{⌀}$$\underset{\to }{k_{12}}$ AA $\underset{\to }{k_{13}}$ ⌀.

This model has two enantiomeric pairs, (^1^R_D_, ^1^R_L_) and (^2^R_D_, ^2^R_L_). This means, according to the previous definitions, that it is a pseudochiral network of order 2. The analysis of the model, using the developed algorithm, confirms its ability to generate SMSB. However, it is important to remark that the outputs obtained with Listanalchem did not always produce SMSB immediately. In those cases, bifurcation diagrams, around the computed values, allowed us to compute the range of values that produce the symmetric breaking. Those bifurcation diagrams were done using Chemkinlator [24]. Figure 1 presents a typical example of the described situation, including three bifurcation diagrams. The right values are easy to find from the values given by the algorithm; for example, a fast way to find SMSB is by exploring the velocity of the entrance of A ($\bar{⌀}$
$\underset{\to }{k_{12}}$ A), see Figure 1B. In this way, it is easy to tune the region of SMSB.

### 8.2. The APED Model

Plasson et al. [25] have proposed the noncatalytic model presented below.

L $\underset{\to }{a}$ L^*^LL $\underset{\to }{h}$ L + LD $\underset{\to }{a}$ D^*^DL $\underset{\to }{h}$ L + DL^*^
$\underset{\to }{b}$ LLD $\underset{\to }{h}$ L + DD^*^
$\underset{\to }{b}$ DDD $\underset{\to }{h}$ D + DL^*^ + L $\underset{\to }{p}$ LLLD $\underset{\to }{e}$ DDD^*^ + L $\underset{\to }{\alpha p}$ DLDD $\underset{\to }{\gamma e}$ LDL^*^ + D $\underset{\to }{\alpha p}$ LDLL $\underset{\to }{\gamma e}$ DLD^*^ + D $\underset{\to }{p}$ DDDL $\underset{\to }{e}$ LL

Our algorithm and software can find values of the rate constants for which the APED model exhibits the breaking of mirror symmetry. We could compute those values without the need for any additional work, as was necessary with the previous model.

Additionally, we would like to remark that the approach developed here to obtain the instability regions has a solid mathematical background which makes it more efficient and general than the brute-force approach used in the original paper of Plasson et al. [25]: a systematic scan of the rate constants. Figure 2 shows an example of the results given by the algorithm.

It is interesting to remark that Plasson et al. [25] claimed that each one of the three sets of reactions:Polymerization: L^*^ + L $\underset{\to }{p}$ LL, D^*^ + L $\underset{\to }{\alpha p}$ DL, L^*^ + D $\underset{\to }{\alpha p}$ LD, and D^*^ + D $\underset{\to }{p}$ DD,Depolymerization: LL $\underset{\to }{h}$ L + L, DL $\underset{\to }{\beta h}$ L + D, LD $\underset{\to }{\beta h}$ L + D,and DD $\underset{\to }{h}$ D + D, andEpimerization: LD $\underset{\to }{e}$ DD, DD $\underset{\to }{\gamma e}$ LD, LL $\underset{\to }{\gamma e}$ DL, and DL $\underset{\to }{e}$ LL,
must have different rate constants, and because of this, they introduce the parameters α, β and γ, which are bigger than zero and different from 1. We found that those constraints are necessary because given α=β=γ=1, the algorithm finds that the SMSB is not possible. Observe that under the equivalent condition, see the legend of Figure 1, the previous model (Replicator Hochberg-Ribo) exhibited SMSB.

### 8.3. The Iwamoto Model

Iwamoto proposed a reaction model including Michaelis-Menten type catalytic reactions [26]. The Iwamoto model is presented below, with perfect and imperfect conditions that depend on the stereoselectivity (R1, R1a, R2, and R2a), and stereospecificity (R3, R3a, R4, and R4a).

PerfectImperfectRP $\overset{k_{1}}{\underset{k_{0}}{⇌}}$ AP $\overset{k_{1}}{\underset{k_{0}}{⇌}}$ AR0A + L $\overset{k_{3}}{\underset{k_{2}}{⇌}}$ 2LA + L $\overset{k_{3}}{\underset{k_{2}}{⇌}}$ 2LR1
A + L $\overset{k_{3a}}{\underset{k_{2a}}{⇌}}$ L + DR1aA + D $\overset{k_{5}}{\underset{k_{4}}{⇌}}$ 2DA + D $\overset{k_{5}}{\underset{k_{4}}{⇌}}$ 2DR2
A + D $\overset{k_{5a}}{\underset{k_{4a}}{⇌}}$ D + LR2aL + E_L_
$\overset{k_{7}}{\underset{k_{6}}{⇌}}$ Z_L_L + E_L_
$\overset{k_{7}}{\underset{k_{6}}{⇌}}$ Z_L_R3
L + E_D_
$\overset{k_{7a}}{\underset{k_{6a}}{⇌}}$ Y_D_R3aD + E_D_
$\overset{k_{9}}{\underset{k_{8}}{⇌}}$ Z_D_D + E_D_
$\overset{k_{9}}{\underset{k_{8}}{⇌}}$ Z_D_R4
D + E_L_
$\overset{k_{9a}}{\underset{k_{8a}}{⇌}}$ Y_L_R4aZ_L_
$\overset{k_{11}}{\underset{k_{10}}{⇌}}$ E_L_ + QZ_L_
$\overset{k_{11}}{\underset{k_{10}}{⇌}}$ E_L_ + QR5
Y_L_
$\overset{k_{11a}}{\underset{k_{10a}}{⇌}}$ E_L_ + QR5aZ_D_
$\overset{k_{13}}{\underset{k_{12}}{⇌}}$ E_D_ + QZ_D_
$\overset{k_{13}}{\underset{k_{12}}{⇌}}$ E_D_ + QR6
Y_D_
$\overset{k_{13a}}{\underset{k_{12a}}{⇌}}$ E_D_ + QR6a

Iwamoto considered to be variables only the species A, L, D, E_L_ and E_D_. The Iwamoto model under perfect conditions shows SMSB. The instability region is a tiny one as shown by Figure 3. However, the algorithm could sample this region without problem.

On the other hand, the Iwamoto model under imperfect conditions is stable if the stereoselective (R1, R1a, R2, and R2a) rate constants satisfy the equalities $k_{2}=k_{2a}=k_{4}=k_{4a}$ and $k_{3}=k_{3a}=k_{5}=k_{5a}$, and at the same time the stereospecific (R3, R3a, R4, and R4a) rate constants satisfy $k_{6}=k_{6a}=k_{8}=k_{8a}$ and $k_{7}=k_{7a}=k_{9}=k_{9a}$. However, if those rate constants are different, as the author assumes, the model presents SMSB, which means: $k_{2}=k_{4}\neq k_{2a}=k_{4a}$, $k_{3}=k_{5}\neq k_{3a}=k_{5a}$, $k_{6}=k_{8}\neq k_{6a}=k_{8a}$ and $k_{7}=k_{9}\neq k_{7a}=k_{9a}$. This latter condition (different rate constants for particular sets of reactions), seems to be necessary to obtain the desired results in those particular models, but it is not clear that it is a condition that can be presupposed of the prebiotic environment. The results for the Iwamoto model under imperfect conditions can be seen in the Appendix A and in the output of Listanalchem.

## 9. Discussion

The developed algorithm and the implemented software are powerful tools, capable of predicting the stability (instability) of models proposed to explain the emergence of homochirality in biological systems. The solid mathematics, behind the algorithm, makes it a robust tool to find the instability regions of chemical reaction mechanisms (networks models) of biological homochirality. This tool can be used to establish the particular rate constant values (intervals), for which a model breaks the initial racemic mixtures. The algorithm and the software can be used to generate phase diagrams of the models proposed to explain the origin of homochirality. Using the previous information, one could study the structure of those models proposed to explain the origin of homochirality in prebiotic earth.

It is important to remark, at this point, that we have focused on the homochiral dynamics that can be triggered by tiny perturbations of the racemic states. Enantiopure states can also be reached by alternate routes, like, for instance, the dynamics that are triggered by large perturbations of those states. We did not study the effect of large perturbations, given that we do not count with the required mathematical tools.

## Figures and Tables

**Figure 1 life-09-00074-f001:**
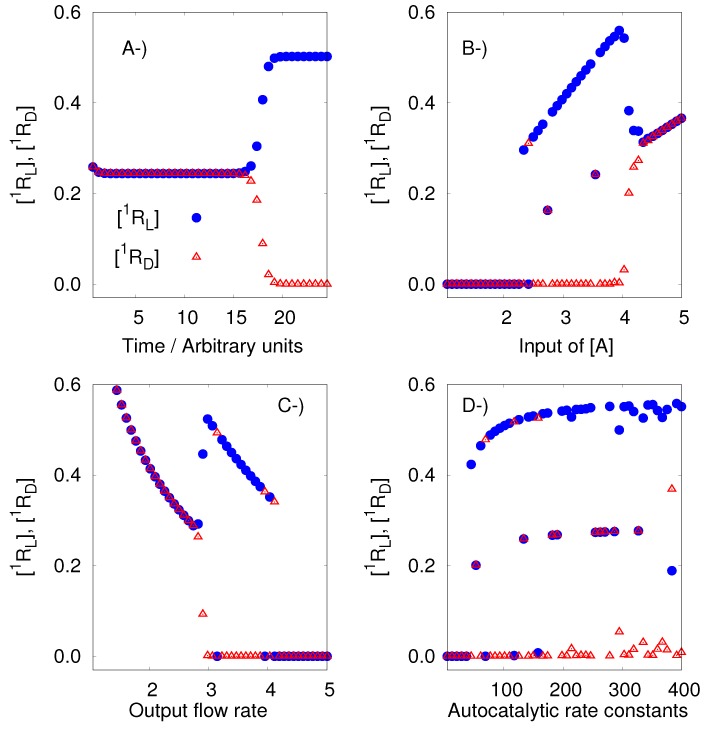
The replicator model of Hochberg and Ribo [27]. A typical simulation result with the kinetic rate constants (and flows) sampled by the algorithm developed in this work. (**A**) Time series; and bifurcation diagrams for (**B**) the input flow rate of A: $\bar{⌀}\;\underset{\to }{k_{12}}A$, (**C**)the output flow rates: ${}^{1}R_{D}\;\underset{\to }{k_{8}}\;⌀$, ^2^R_D_
$\underset{\to }{k_{9}}$ ⌀, ^1^R_L_
$\underset{\to }{k_{10}}$ ⌀, ^2^R_L_
$\underset{\to }{k_{11}}$ ⌀, and A $\underset{\to }{k_{13}}$ ⌀, and (**D**) the autocatalytic rate constants: A + ^1^R_D_ + ^2^R_D_
$\underset{\to }{k_{0}}$ 2^1^R_D_ + ^2^R_D_, A + ^1^R_L_ + ^2^R_L_
$\underset{\to }{k_{2}}$ 2^1^R_L_ + ^2^R_L_, A + ^2^R_D_ + ^1^R_D_
$\underset{\to }{k_{4}}$ 2^2^R_D_ + ^1^RD, and A + ^2^R_L_ + ^1^R_L_
$\underset{\to }{k_{6}}$ 2^2^R_L_ + ^1^R_L_. The full set of rate constant, for this particular simulation, can be seen in the output of Listanalchem presented in the Appendix A. However, it is important to emphasize that $k_{0}=k_{2}=k_{4}=k_{6}$, $k_{1}=k_{3}=k_{5}=k_{7}$, and $k_{8}=k_{9}=k_{10}=k_{11}=k_{13}$. Also, an initial enantiomeric excess, $ee=\frac{\left[{}^{1}R_{L}\right]-\left[{}^{1}R_{D}\right]}{\left[{}^{1}R_{L}\right]+\left[{}^{1}R_{D}\right]}=4.207852\times 10^{-16}$ was used.

**Figure 2 life-09-00074-f002:**
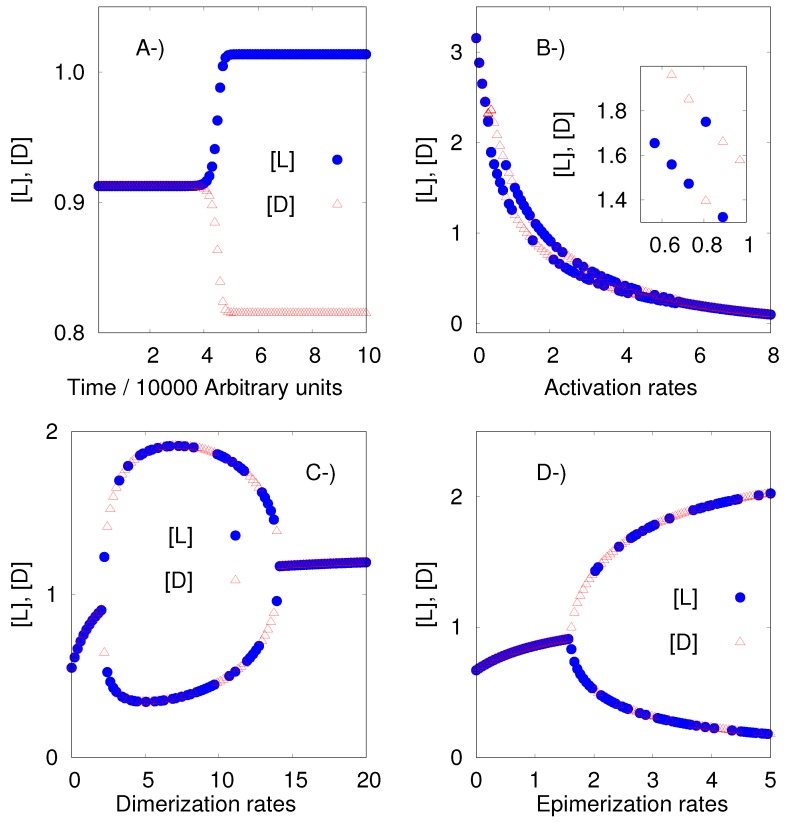
The APED model of Plasson et al. [25]. (**A**) Time series; and bifurcation diagrams for (**B**) the activation rate constants of reactions L $\underset{\to }{a}$ L^*^ and D $\underset{\to }{a}$ D^*^, (**C**) the dimerization rate constants of reactions L^*^ + L $\underset{\to }{p}$ LL and D^*^ + D $\underset{\to }{p}$ DD, and (**D**) the epimerization rate constants of reactions LD $\underset{\to }{e}$ DD and DL $\underset{\to }{e}$ LL. In this case, the initial concentrations of the enantiomers were taken equal, the numerical error of the computer calculations was enough to break the initial racemic mixture.

**Figure 3 life-09-00074-f003:**
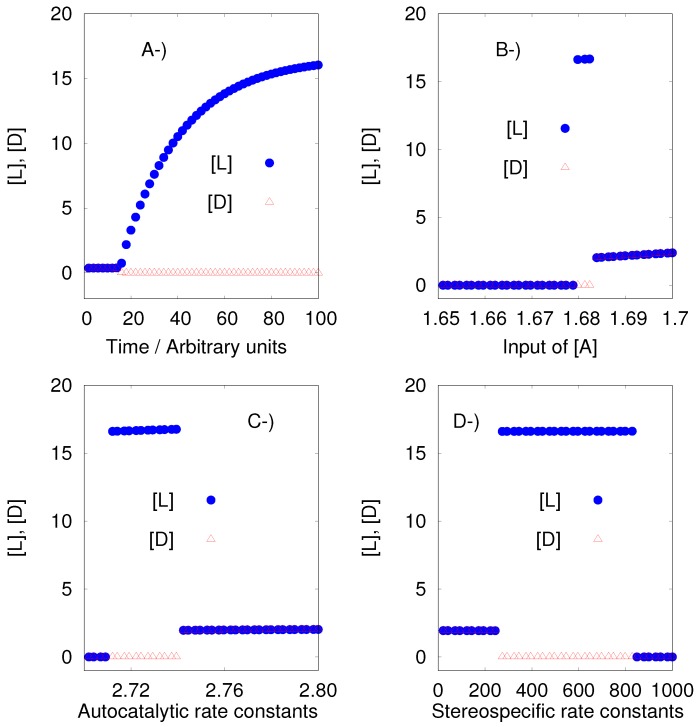
The Iwamoto model under perfect conditions [26]. (**A**) Time series; and bifurcation diagrams for (**B**) input flow reaction P $\underset{\to }{k_{0}}$ A, (**C**) autocatalytic reactions A + L $\underset{\to }{k_{2}}$ 2L and A + D $\underset{\to }{k_{2}}$ 2D, and (**D**) stereospecific reactions L + EL $\underset{\to }{k_{6}}$ ZL and D + ED $\underset{\to }{k_{8}}$ ZD. The initial concentrations of the enantiomers L and D were equal until the 11th decimal position, equivalent to an enantiomeric excess of, $ee=\frac{\left[L\right]-\left[D\right]}{\left[L\right]+\left[D\right]}$, $2.217111\times 10^{-12}$.

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
