# Peer review of "Computing the Parameter Values for the Emergence of Homochirality in Complex Networks"

_life, 2019, doi:10.3390/life9030074_

Round 1

Reviewer 1 Report

See accompanying file.

Reviewer 2 Report

This paper describes an appealing idea to facilitate discerning those chemical reaction networks which are potentially capable, i.e. in purely mathematical terms, to generate chiral symmetry breaking steady states. The certainly useful mathematical idea of the authors is to exploit observed symmetries of the Jacobian in stability analysis of these reaction networks. Further, the stoichiometric network analysis is employed to simplify the problem for more complex cases.

Frank's seminal mathematical example of a reaction network capable of absolute asymmetric synthesis serves as an entry point. Other networks studied are those of Hochberg and Ribo, of Plasson, Iwamoto and of Calvin.

The analysis is sound and the paper well-written in a lucid style. The mathematics is well-developed in an easily comprehensible way.

I have nonetheless some suggestions for improvement and also some more serious remarks, which should be addressed before this paper can be published:

Remark 3 (p. 2) and p.4 (line 117) , p.3 (line 83), p.6 (line 169): "enantioselective synthesis" in chemistry is actually the stereoselective process under the influence of a chiral catalyst! What the authors here actually mean - and which they should write to avoid confusion - is precisely "absolute asymmetric synthesis" (which is even more general than the also correct "asymmetric autocatalysis", as the latter would not necessarily encompass e.g. the APED model). That term was also used by Frank.

Definition 5 on p.3: Are the authors sure that all reaction rates always vanish identically at the steady state (as opposed to the chemical equilibrium)? Maybe it would be better to specify that this holds only for the set of forward reaction rates alone and that there could also be complex networks which contain loops, i.e. cyclic processes where not even all forward rates vanish identically at the steady state (not to be confused with the Onsager triangle reaction).

I strongly question eq. 11 on p.3: 0 = - [L] - [D] means [L] + [D] = 0, i.e. the sum of concentrations of chiral matter is zero at the steady state!? What kind of steady state is this supposed to be ? A racemic steady state is actually characterized by [L] = [D] for, of course, non-zero concentrations! Why should the rate d[A]/dt vanish at all at a steady state (see eq. 9)? - unless the reaction would have gone to completion in a closed matter system (in which case the concentrations of [L] + [D] = [A]initial (with [A]initial being the concentration of A at time zero). The authors are advised to sharpen their definition of what a "steady state" actually is (the same holds for my second remark above). Please note that a physical realization of Frank's model implies a system open to matter flow.

p.4 (line 115): "enantiomeric pairs cannot be distinguished by electromagnetic forces". This is plainly wrong, otherwise there would be no "optical activity" of chiral molecules which goes back to purely electromagnetic interaction of the molecule's electron cloud with the E or B fields of a light wave. And it implies (so the authors proceed in line 116): "that those two species react with the same chemical species at the same reaction rates". This is true only if the other chemical species is achiral! And the indiscernibility of enantiomers is also only with respect to scalar and uniform (more precisely: time-invariant) forces. Please correct.

In the introduction, the following paper should be cited, in which it was already investigated mathematically a hypothetical reversible Frank mechanism through (linear) stability analysis to obtain (for this special case) mathematical conditions for symmetry breaking and generation of homochiral states: ChemPhysChem 2008, 9 (16), 2359-2371.

There are some unnecessary spelling mistakes, e.g. it should always be "contraint"/"constraints" and not "constrains". On p. 7 (line 210) it should read "not produce" instead of "no produce". p.11 (line 326): "become unstable" instead of "becomes unstable", same for p. 15 (line 430). p. 17 (line 498): "first of all" instead of "first at all". p. 18 (Figure caption): "The full set of rate constant" instead of "The full set of rates constants". Same page, lines 524 and 525: write "Polymerization" instead of "Polimerization", same for "Depolimerization". p 19 (line 528): "different from 1" instead of "different than 1". p. 20 (line 538): "as it is shown" instead of "as it is showed". Same page, line 555: write "rate constant values" instead of "rate constants values".
